
# 1 Single Point Positioning with Vertical Total Electron Content
# 2 estimation based on single epoch data

Artur Fischer[1], Sławomir Cellmer[1], and Krzysztof Nowel[1]
[1]Department of Geodesy, University of Warmia and Mazury, Olsztyn, 10-719, Poland
*Correspondence to*: Artur Fischer (artur.fischer21@gmail.com)
**Abstract.** This paper proposes a new mathematical method of ionospheric delay estimation in single point positioning (SPP)
using a single-frequency receiver. The proposed approach focuses on the $\Delta VTEC$ component estimation (MSPPwithdVTEC)
with the assumption of an initial and constant value equal to 5 in any observed epoch. The principal purpose of the study is
to examine the reliability of this approach to become independent from the external data in the ionospheric correction
calculation process. To verify the **MSPPwithdVTEC**, the SPP with the Klobuchar algorithm was employed as a reference
model, utilizing the coefficients from the navigation message. Moreover, to specify the level of precision of the
**MSPPwithdVTEC**, the SPP with the IGS TEC map was adopted for comparison as the high-quality product in the
ionospheric delay determination. To perform the computational tests, real code data was involved from three different
localizations in Scandinavia using two parallel days. The criterion were the ionospheric changes depending on geodetic
latitude. Referring to the Klobuchar model, the **MSPPwithdVTEC** obtained a significant improvement of 15 – 25% in the
final SPP solutions. For the SPP approach employing the IGS TEC map and for the **MSPPwithdVTEC**, the difference in
error reduction was not significant, and it did not exceed 1.0% for the IGS TEC map. Therefore, the **MSPPwithdVTEC** can
be assessed as an accurate SPP method based on error reduction value, close to the SPP approach with the IGS TEC map.
The main advantage of the proposed approach is that it does not need external data.

## 20 1 Introduction

Single point positioning (SPP) allows of the indication of an autonomous position of a receiver using code data from
the Global Positioning System (GPS). Code ranges are not ambiguous and do not require to apply the precise method of
ambiguity initialization (Bakuła, 2020). The principal problem of SPP stems from different types of errors degrading the
GPS signal between a rover and a specified satellite in a given epoch. Ionospheric delay contributes to the general GPS error
budget by its volatility in the range of 40 – 60 m during daytime and 6 – 12 m at night (US Army Corps of Engineers, 2003).
The ionosphere consists of charged particles that appear because of the ionization process (El-Rabbany, 2002; Awange,
2012). Problems with ionosphere modeling come from difficulties between solar activity and the geomagnetic field
interactions (Xu and Xu, 2016). The basic concepts of the GPS signals delay were briefly considered by Golubkov et al. and
Kuverova et al. (Golubkov et al., 2018; Kuverova et al., 2018; Golubkov et al., 2019). To specify a suitable magnitude of


delayed GPS signal along an appropriate path between satellite and receiver, a proportional quantity such as Total Electron
Content (TEC) has to be involved and defined as the linear integral of the density of the particles alongside the ray path
(Cooper et al., 2019). The TEC unit is equal to $10^{16}$ electrons per square meter (in the cross-section of 1 m$^2$) (Ciraolo, 2005).
To calculate and reduce such effect on the GPS code measurement, Stępniak (2016) distinguished different types of models
and mathematical estimating methods: physical - theoretical (e.g. Chapman's model), physical - empirical (e.g. IRI and the
NeQuick model), mathematical - deterministic (based on a mathematics function), and mathematical – stochastic (based on a
large set of processed data used to describe the spatial-temporal changes of ionosphere) e.g. the IGS model.

The authors propose the autonomous SPP approach with $\Delta VTEC$ component estimation using single-frequency GPS

code observations to be independent of external products, e.g. an IGS TEC map. The disadvantage of the mathematical
models is performing an ionospheric effect calculation mostly in post-processing. Since many mathematical approaches to
self-sufficient ionospheric delay modeling have been proposed, especially in the carrier phase domain using multi-frequency
observations, the authors wanted to introduce a new estimation method employing single-frequency GPS code observations.
For instance, Georgiadiou (1994) proposed a mathematical method based on differences between the pseudo-ranges
measured on the L1 and L2 carries frequency, respectively (dual-frequency method). The computational tests with
comparison to the reference method without ionospheric corrections were done by Camargo et al. (2000), focusing
particularly on the pseudo-ranges filtered by the carrier phase. The method of slant delay estimation (STEC – alongside a
line of sight) in the L1 carrier reduced 80% of errors related to ionospheric effects in the point positioning technique, also
delivering improvement solutions during the ionosphere maximum. Bosy (2005) described a geometry-free linear
combination which can be employed to ionosphere modeling, with simultaneous consideration and repair of cycle-slip
effects and other parameters of GPS vector - ambiguity and tropospheric effects. Krypiak-Gregorczyk and Wielgosz (2018)
proposed the use of multi-frequency GNSS signals for TEC modeling, utilizing the carrier phase bias of a geometry-free
linear combination. The received bias accuracy results on the level of 7 – 8 cm allow TEC computation with desirable
uncertainty, i.e. lower than 1 TECU. Additionally, an ionosphere-free linear combination as an independent positioning
approach can also be well adapted to minimize the ionosphere negative impact on GPS positioning (Teunissen and
Kleusberg, 1998). However, Hofmann-Wellenhof et al. (2008) stated that "ionosphere-free" is not an entirely correct name,
caused by the approximation existing in the process of making the refractive index. Those authors studied an ionosphere-free
approach in the code SPP and achieved a beneficial magnitude of error reduction (50-60%) in relation to the reference SPP
model without ionospheric corrections.

On the contrary, empirical models do not significantly reduce the ionosphere influence in the GPS positioning as

mathematical (deterministic) methods, but can make real-time improvements by using the external data, e.g. coefficients
transmitted in the navigation message to correct the signal pseudo-ranges. One of these is the Klobuchar algorithm (see
Klobuchar, 1987), which compensates for 50 – 60% of the ionospheric range error, utilizing a single-layer model of the
ionosphere (Leick et al., 2015). In the current study, the authors wanted to treat the SPP method with the Klobuchar
algorithm as a reference method, because of its popularity and utility in GPS measurement. A significant improvement can


be noted in the vertical component which is the most affected by the atmospheric delay. Júnior et al. (2019) investigated the
analysis of the Klobuchar model in the ionospheric delay reduction procedure utilizing code observation in point positioning.
The algorithm works clearly when ionosphere activity is significant and improves vertical solutions by 67%. For the
horizontal components, the improvement using the Klobuchar algorithm is up to 9% regarding the non-iono model. It should
be noted that GPS point positioning using the Klobuchar algorithm can degrade the position because of the constant value of
the ionospheric delay (up to 5 ns SET) during nighttime.

High-quality representation of the ionosphere influence on positioning can be obtained by the Global Ionospheric
Models (GIMs), used mostly in the post-processing purposes as explained in Ciećko and Grunwald (2020). It is worth noting
that Abdelazeem et al. (2016) developed the regional ionospheric model over the European area and implemented it in
Precise Point Positioning (PPP), operating in real-time using the real-time service products (RTS) of the International GNSS
Service (IGS). The results present an improvement in the accuracy on the level of 40 % (under the mid-latitude region) in the
3D position relating to the IGS-GIM. The accuracy is higher primarily because of the better temporal and spatial resolution
of the model (15' and 1° x 1°), while the IGS TEC map includes nodes containing the appropriate VTEC value with a time
resolution of 1 hour and a spatial resolution of 2.5° x 5°, respectively for latitude and longitude. In turn, Krypiak-Gregorczyk
et al. (2017) prepared the ionosphere model covering the Europe region as well, based on multi-GNSS data. The solutions
are beneficial because they have 2-3 times lower RMS value than the results of GIMs, e.g. from IGS. Zhang et al. (2019)
also examined global ionospheric maps operating in real-time, dedicated to single-frequency positioning. Chen and Gao
(2005) tested the IGS TEC map as the basic condition to assess the precision of the PPP model using different procedures to
resolve the ionospheric delay problem such as single-frequency ionosphere-free linear combination (averages un-differenced
code and carrier-phase observations on the same frequency) or estimation of the ionospheric effect as an unknown
parameter. The advantage of the methods is no need for external products. For instance, the estimation method achieved
comparable accuracy in the mid-latitude stations but for the higher latitude, the GIM is still quite better, inversely on the
equatorial stations. This encourages a focus on the IGS TEC map as the high accuracy product to authenticate solutions from
the suggested approach to SPP, and to validate the autonomous method of the ionospheric delay calculation. It should be
noted that although the efficiency of GIMs is not significant using GPS code observations, the accuracy is suitable enough
for navigation goals and further development of this concept.

In sum, the motivation of this paper is to analyze a new mathematical method of ionospheric delay estimation to
improve the SPP. The authors put forward the hypothesis to be independent of external data use in the meaning of the new
method in the ionospheric delay calculation procedure.
**2 SPP mathematical models**

In this section, the grounds of the common used SPP mathematical models using Klobuchar algorithm and IGS TEC
map will be introduced, and the proposition of the new strategy of SPP determination by use of simple as well as


autonomous method to estimate the ionospheric delay. This is followed by the appropriate algorithm presentations with
suitable explanations. In addition, the accuracy analysis criteria will be described in view of models credibility procedure.

**2.1 SPP with ionospheric corrections using Klobuchar algorithm and IGS TEC map**

In this study, the Klobuchar model was adapted as a reference in the SPP accuracy tests. Eight model coefficients
transmitted via navigation message are the primary components involved in the algorithm to reduce the ionosphere effect in
the SPP. The geodetic coordinates of the GPS antenna, GPS observing time (in seconds) as well as azimuth and elevation of
observed satellites as viewed from the receiver are needed to be known. The formula to calculate the ionospheric correction
based on the Klobuchar algorithm is as follows (Hofmann-Wellenhof et al., 2008):

$$\Delta T_V^{Iono} = A_1 + A_2 \cos\left(\frac{2\pi(t-A_3)}{A_4}\right) \tag{1}$$

where $A_1$ is a constant value of 5 ns. In turn, $A_2$ is a sum of multiplying four α coefficients and the geomagnetic latitude of
an ionospheric pierce point $\varphi_{IP}^m$. $t$ means GPS time of the ionospheric pierce point. $A_3$ is 14:00 local time which specifies the
highest ionospheric disturbance. $A_4$ means the same as $A_3$ but there are four β coefficients are multiplied by $\varphi_{IP}^m$.
To obtain an ionospheric delay alongside the GPS signal travel path, the mapping function should be employed. Thus,
the concept of the ionospheric point has to be expanded as a piercing point of the GPS wave path and the
ionospheric single layer on the specified altitude. Thus, the zenith angle at the piercing point should first be indicated
(Hofmann-Wellenhof et al., 2008):

$$\sin z' = \frac{R_e}{R_e + h_m} \sin z_0 \tag{2}$$

$R_e$ is Earth radius 6370 km and $z_0$ means a zenith angle from the receiver site. $h_m$ is defined as the height of the ionospheric
pierce point. In general, $h_m$ is identified by the single-layer model where all free electrons are concentrated in the infinitesimal
spherical shell at the assumed altitude - 450 km. Other formulations are possible too, for instance, from the Klobuchar
algorithm, presented in Rui et al. (2011):

$$mF = 1 + 16 \cdot \left(0.53 - \frac{E}{\pi}\right)^3 \tag{3}$$

where $E$ means an elevation angle in the slant factor calculation.
It should be also noted that the type of mapping function in the atmospheric effect calculation process contributes to the
final solution accuracy as well. Allain et al. (2009) examined the tomographic mapping function known as Multi-Instrument
Data Analysis System (MIDAS) to ionospheric effect determination for the single-frequency data. Research has shown that
daily positioning errors are up to 50% lower in comparison to positioning using the Klobuchar algorithm or International


Reference Ionosphere (IRI) when the surrounding distribution of receivers is favorable. Regardless of the map type, dual-
frequency observations allow for even greater precision of the ionospheric effect mitigation in the GPS pseudo-range
measurement.

Therefore, the mapping function can be used as an inverse of the cosine function (Hofmann-Wellenhof et al., 2008):

$$\Delta T_s^{Iono} = \Delta T_v^{Iono} / \cos z' \tag{4}$$

Finally, the ionospheric delay alongside the rover-satellite is achieved in seconds. To obtain the metric magnitude of
the calculated effect, $\Delta T_s^{Iono}$ is multiplied by the speed of light. The Klobuchar algorithm was fully described by Xu (2007).
To future elaboration, $\Delta T_s^{Iono}$ will be denoted as $\delta_K$ where subscript is appropriate for the Klobuchar method.

The second approach is SPP with ionospheric corrections computed based on the IGS TEC map. This method is used to
examine and verify the quality of the new autonomous, estimation method of the ionospheric effect in the SPP.
Consequently, ionospheric delay as the base formula in the zenith direction can be introduced (Schüler, 2001):
$$\delta_{IT} = \int_{h_m}^{\infty} \frac{C_2}{f^2} = \frac{C}{f^2} \int_{h_m}^{\infty} N_e(h) \cdot dh = \frac{C}{f^2} \cdot VTEC \tag{5}$$
where the subscript is proper for the IGS TEC map product. $C$ is a constant value of 40.3 m³/s², $f$ is an appropriate frequency,
and $VTEC$ is naturally the vertical total electron content in TECU units. $N_e$ is electron density factor [electrons/m³], and $h$ is
equal to the travelled ray path from the satellite to the rover. In turn, $h_m$ is the height of the single layer of the ionosphere or
height of the piercing point for which the appropriate VTEC value from IGS TEC is interpolating. Hence, there is a need to
indicate the geodetic coordinates for ionospheric pierce point using e.g. geometric method formulation (Prol et al., 2017).
Taking into account ionospheric delay as a proportional value to TEC and proportional to the distance covered across
the band, the relation of VTEC and TEC can be defined (Leick et al., 2015):
$$VTEC = \cos z' \cdot TEC \tag{6}$$
To integrate VTEC to STEC, the ionospheric mapping function, mentioned in the Eq. (2) is presented as an inverse of the
cosines function (Leick et al., 2015):
$$F(z') = \frac{1}{\cos z'} = \left[ 1 - \left( \frac{R_e \sin z'}{R_e + h_m} \right)^2 \right]^{-\frac{1}{2}} \tag{7}$$
where the adopted $z'$ angle is equivalent to the zenith angle at the piercing point in (4).
Using Eq. (5), (6), and (7), the ionospheric correction can be obtained in the ray path direction between satellite-rover:



$$\delta_{IT} = \frac{40.3}{f^2} \cdot F(z') \cdot VTEC \qquad (8)$$
Therefore, to briefly explain the mathematical model of SPP with utilized ionospheric corrections, the code observation
equation was adapted based on Strang and Borre (2008) with complementary changes:
$$\begin{cases} P_r^s = \rho_r^s + c(\Delta t_r - \Delta t^s) + \delta_{TROP} + \delta_K + \varepsilon_P \\ P_r^s = \rho_r^s + c(\Delta t_r - \Delta t^s) + \delta_{TROP} + \delta_{IT} + \varepsilon_P \end{cases} \qquad (9)$$
where the first equation is concerning on the SPP approach with Klobuchar algorithm and the second one is referring
to the IGS TEC map. The left side is the measured pseudo-range. On the right side are the model and estimated magnitudes:
the geometrical distance between rover and satellite (satellite coordinates computed by utilization of the ephemeris
information – SP3 file), speed of light, receiver and satellite clock biases, tropospheric delay, ionospheric delay (computed
using Klobuchar algorithm (eight coefficients from navigation message) or IGS TEC map utilizing IONEX file) and pseudo-
range remaining error, respectively. In the research, the tropospheric corrections were obtained based on Hopfield (see
Hopfield, 1969) using model values of the dry and the wet subcomponents. Additionally, the clock bias of satellites has been
received by the utilization of satellites' ephemeris data and the relativistic improvements.

### 2.2 Modified SPP with autonomous VTEC estimation method

The essence of the proposed modified SPP method lies in an estimation of the $\Delta VTEC$ term which is a variable
component of the ionospheric delay:
$$\delta_{IONest} = \frac{40.3 \cdot 10^{16}}{f^2} \cdot F(z') \cdot (VTEC_0 + \Delta VTEC) \qquad (10)$$
the frequency in the real tests was adopted as the L1 carrier of the GPS signal: 1575.42 MHz.
The modified SPP model with an independent method of the ionospheric effect estimation is expressed in the system of
equations:
$$\begin{cases} P_1 = \rho_r^{s_1} + c(\Delta t_r - \Delta t^{s_1}) + \delta_{TROP_1} + \delta_{IONest_1} + \varepsilon_1 \\ P_2 = \rho_r^{s_2} + c(\Delta t_r - \Delta t^{s_2}) + \delta_{TROP_2} + \delta_{IONest_2} + \varepsilon_2 \\ \quad . \\ \quad . \\ \quad . \\ P_n = \rho_r^{s_n} + c(\Delta t_r - \Delta t^{s_n}) + \delta_{TROP_n} + \delta_{IONest_n} + \varepsilon_n \\ VTEC^{pseudoobs} = VTEC_0 + \Delta VTEC + \varepsilon_{\Delta VTEC} \end{cases} \qquad (11)$$





The last row is a pseudo-observation equation in which $VTEC_0$ is the constant, initial value of VTECs in a given epoch,
appropriate for all satellite elevation, $\Delta VTEC$ is an estimated ingredient and $\varepsilon_{\Delta VTEC}$ is a remaining error of determining factor. It
was decided, after performing many tests, to include this pseudo-observation equation into the SPP approach to ensure a
stable GPS solution. The model without the pseudo-observation formula would be too weak to give stable results (note that
single epoch positioning is used).
After many computational tests, it was assumed that the initial value of $VTEC_0$ in any measured epochs during daytime
and nighttime of SPP is **5 TECU.** Therefore, the method does not need external information about VTEC referring to the
piercing point on the line of sight receiver – satellite, even if the IGS TEC map is available, it indicates that the model is
simple to build and implement into a complex algorithm. The reliability and usefulness will be submitted during the
presentation of the results.
It is assumed in this method that the "observed" and approximate values are equal:
$$VTEC^{pseudoobs} = VTEC_0 \tag{12}$$
Continuing, to simplify successive descriptions of the modified SPP approach, the mapping coefficient is denoted:
$$mapcoeff = \frac{40.3 \cdot 10^{16}}{f^2} F(z') \tag{13}$$
The system of code equations (11) after linearization can be introduced in the matrix notation:
$$\mathbf{V} = \mathbf{AX} - \mathbf{L} \tag{14}$$
where:
$$\mathbf{V} = \begin{bmatrix} -\varepsilon_1 \\ \vdots \\ -\varepsilon_n \end{bmatrix} \tag{15}$$
is a residual vector,
$$\mathbf{A} = \begin{bmatrix} a_{11} & a_{12} & a_{13} & 1 & \vdots & mapcoeff^1 \\ \vdots & \vdots & \vdots & \vdots & \vdots & \vdots \\ a_{n1} & a_{n2} & a_{n3} & 1 & \vdots & mapcoeff^n \\ \hdashline 0 & 0 & 0 & 0 & \vdots & 1 \end{bmatrix} \tag{16}$$
is a design matrix.
The vector of unknowns receives an additional parameter in the adjustment process:





$\mathbf{X} = \begin{bmatrix} \Delta X_r \\ \Delta Y_r \\ \Delta Z_r \\ c\Delta t_r \\ \hline \Delta VTEC \end{bmatrix}$ (17)
The disclosure vector is:
$\mathbf{L} = \begin{bmatrix} L_r^1 \\ \vdots \\ L_r^n \\ \hline 0 \end{bmatrix}$ (18)
where $L_r^i = P_i - \rho_i^r + c\Delta t^i - \delta_{TROP_i} - mapcoeff \cdot VTEC_0$. The last entry amounts to zero because of assumption (12).
The weight matrix has been prepared based on pseudo-range measurement error which was assumed as a 2.00 m and
appropriate satellite elevation angle. The criterion of the minimal mask was implemented as a 10 degree. After
computational tests with theoretical analysis, the weight of the estimated component $\Delta VTEC$ was assumed in the model as 1.
$\mathbf{P} = \begin{bmatrix} \dfrac{1}{\delta^2}\sin(elev_1) & \cdots & 0 & 0 \\ \vdots & \ddots & \vdots & \vdots \\ 0 & \cdots & \dfrac{1}{\delta^2}\sin(elev_n) & 0 \\ \hline 0 & \cdots & 0 & 1 \end{bmatrix}$ (19)
The least-squares estimate of the Eq. (14) is computed from the normal equations:
$\mathbf{A^T PA\hat{X} - A^T PL = 0}$ (20)
together with its covariance matrix:
$\mathbf{C_{\hat{x}} = m_0^2 (A^T PA)^{-1}}$ (21)
with the variance factor: $m_0^2 = \dfrac{\mathbf{V^T PV}}{n-m}$.
The number of parameters $m = 5$. Thus, the minimal number of observations should be $n = 6$ to ensure necessary
redundancy.
**2.3 Accuracy analysis criteria**
The basic statistical operator in the experiment is a distance of the solution from the true position (*dist*) as well as its
average value (*DIST*), computed from solutions obtained from the single epochs with its mean error. The actual position





means constant station coordinates provided by the agency, which manage the Continuously Operating Reference Station
(CORS) used in the experiment for evaluation of the positioning model accuracy. The formula can be introduced in each
epoch as follows:
$$dist_{ep_i} = \sqrt{(X_r - X_t)^2 + (Y_r - Y_t)^2 + (Z_r - Z_t)^2} \qquad (22)$$
where subscript "$r$" means calculated rover's coordinates and "$t$" regarding to the actual position.

Therefore:

$$m^2_{dist_{ep_i}} = \mathbf{W C}_{\hat{X}_{ep_1}} \mathbf{W'} \qquad (23)$$
where $\mathbf{C}_{\hat{X}}$ is a covariance matrix of the parameter vector and $\mathbf{W}$ is a gradient:
$$\mathbf{W} = \left[ \frac{\Delta X_{ep_i}}{dist_{ep_i}} \quad \frac{\Delta Y_{ep_i}}{dist_{ep_i}} \quad \frac{\Delta Z_{ep_i}}{dist_{ep_i}} \right] \qquad (24)$$

The average value is as follows:

$$m^2_{DIST} = \frac{1}{n^2} \sum_{i=1}^{n} m^2_{dist_{ep_i}} \qquad (25)$$

The NEU (North East Up) coordinates system was used in the comparative analysis, where the calculated rover's

position is compared to the actual position. Therefore, the rotation matrix was used to convert the covariance matrix (21) of
the parameters to the NEU system:
$$\mathbf{C_{NEU}} = \mathbf{R C}_{\hat{X}} \mathbf{R}^T \qquad (26)$$
where:
$$\mathbf{R} = \begin{bmatrix} -\sin\varphi\cos\lambda & -\sin\varphi\sin\lambda & \cos\varphi \\ -\sin\lambda & \cos\lambda & 0 \\ \cos\varphi\cos\lambda & \cos\varphi\sin\lambda & \sin\varphi \end{bmatrix} \qquad (27)$$
The $\varphi$ and $\lambda$ are rover geodetic coordinates.

The covariance matrix of mean values computed from the whole observational day is:

$$\mathbf{C_{NEU_{mean}}} = \mathbf{D C_{NEU_{set}}} \mathbf{D^T} = \frac{1}{n^2} \sum_{i=1}^{n} \mathbf{C_{NEU_{ep_1}}} \qquad (28)$$
where $\mathbf{C_{NEU_{set}}}$ is a block matrix which contains on the diagonal the covariance matrixes in the NEU setup from all measured
epochs ($n$) and $\mathbf{D}$ is treated as a transition matrix from the NEU to their mean values:



$$\mathbf{D} = \begin{bmatrix} \dfrac{1}{n} & 0 & 0 & \dfrac{1}{n} & 0 & 0 & \cdots & \dfrac{1}{n} & 0 & 0 \\ 0 & \dfrac{1}{n} & 0 & 0 & \dfrac{1}{n} & 0 & \cdots & 0 & \dfrac{1}{n} & 0 \\ 0 & 0 & \dfrac{1}{n} & 0 & 0 & \dfrac{1}{n} & \cdots & 0 & 0 & \dfrac{1}{n} \end{bmatrix}$$ (29)

## 3 Numerical experiment and discussion

In this section, the explanation of the research concept will be done. Next, the appropriate numerical experiment in
view of graphics and numeric settings. The parallel discussion about obtained results for appropriate interpretation will be
made.

### 3.1 Research concept

The numerical experiment is based on real single frequency code pseudorange observations. Namely, C1C code data on
the L1 carrier frequency (1575.42 MHz). Continuing, three different EURE Permanent GNSS Network stations have been
chosen in Scandinavia. Two stations in Sweden – Visby (VIS) and Skellefteå (SKE), one in Norway – Vardø (VARS). The
observational files and initial coordinates of receivers was gained from the BKG (Bundesamt für Kartographie und
Geodäsie) GNSS Data Center. The parameters of satellite orbits (SP3 file) and atmospheric data were obtained by means of
CDDIS (Crustal Dynamics Data Information System) - in fact, IONEX (IONosphere map EXchange format) only in view of
atmospheric data, as a source of IGS TEC map. The coordinates of points were treated as the true coordinates in the practical
part of the experiment. The reference coordinates are presented in the table:
**Table 1.** Actual coordinates of points

| Points | X | Y | Z |
|---|---|---|---|
| VIS600SWE | 3246466.556 | 1077901.829 | 5365279.606 |
| SKE800SWE | 2534032.877 | 9751679.370 | 5752078.718 |
| VARS00NOR | 1844607.623 | 1109719.107 | 5983936.007 |

In the models, the actual coordinates have been converted to the antenna phase center to make a comparative analysis with
the SPP results, where measurements were executed to the antenna phase center.
Three different localizations allow checking how the modified SPP model works on different geodetic latitude because
of ionosphere activity changes, so its quality in the GPS code domain can be widely stated.
The research concept focuses on measurement on two different days in the cited locations. Therefore, three stations of
the EUREF Permanent GNSS Network were employed for comparative analysis based on data from two parallel days. The
table below presents the structure of the experiment:



**Table 2.** Experiment concept

| Points | Days | SPP methods |
|---|---|---|
| VIS600SWE | 15/06/2019 15/08/2019 | SPPwithKM (SPP with Klobuchar model) MSPPwithdVTEC (Modified SPP with **Δ**VTEC estimation) SPPwithITM (SPP with IGS TEC map) |
| SKE800SWE | 15/06/2019 15/08/2019 | SPPwithKM MSPPwithdVTEC SPPwithITM |
| VARS00NOR | 15/06/2019 15/08/2019 | SPPwithKM MSPPwithdVTEC SPPwithITM |

To execute the practical part of the research, the MATLAB environment from The MathWorks was used. The "PostCalc" software developed by Dawid Kwaśniak was implemented with the complementary changes done by the authors.

**3.2 Discussion of the experiment results**

The Figures 1-3 present the distribution of *dist* values during the observational day (*Results of the positioning models*) and their average value *DIST* with appropriate mean errors in the middle (*Average results of the positioning models*). In turn, the bottom parts show the error reduction of the models (*Differences of the positioning models*). The upper part of Figure 1(**a**) demonstrates the solutions for Visby station on 15 June, 2019. The *dist* results are significantly improved for **MSPPwithdVTEC** referring to the **SPPwithKM** what is confirmed by the average value of *DIST* equalled to 4.886 m. There is not a major difference of *DIST* between **MSPPwithdVTEC** and **SPPwithITM** (0.033 m). Therefore, the mean error of *DIST* (0.072 m) affirms the precision of the modified solution. Studying the bottom division of Figure 1(**a**), **SPPwithKM** was assumed as a reference one (100%) in the calculation of the percent values of error reduction based on *DIST*. The results are satisfying because of error reduction on the level of 22.97% in the **MSPPwithdVTEC** case and the close discrepancy with the error reduction of the **SPPwithITM** (0.53%). The second day using Visby station is 15 August, 2019. In the middle of Figure 1(**b**), *DIST* is beneficial for the **MSPPwithdVTEC** (4.912 m) compared to the reference model which leads to defining the tendency of improved accuracy in the SPP. Again, the difference in the average solutions of *DIST* between **MSPPwithdVTEC** and **SPPwithITM** is insignificant (0.055 m) according to code observations accuracy level. Thus, the accuracy of the estimation method is comparable with the IGS TEC map. Focusing on the average explanation of the *DIST* mean errors among the **MSPPwithdVTEC** (0.067 m) and the **SPPwithITM** (0.074 m), these approaches do not distinctly vary, which indicates that the proposed SPP model works well. In the bottom of Figure 1(**b**), the error reduction of **MSPPwithdVTEC** is 20.90% and is at a similar level with **SPPwithITM** (21.79%). The **SPPwithKM** proved to be the lowest accuracy method. Probably, the ionospheric corrections obtained by the coefficients from the navigation message cannot reflect the changes that take place in the ionosphere with the higher temporal accuracy. Briefly, in the first studied point, the **MSPPwithdVTEC** can be judged as the precise SPP model.

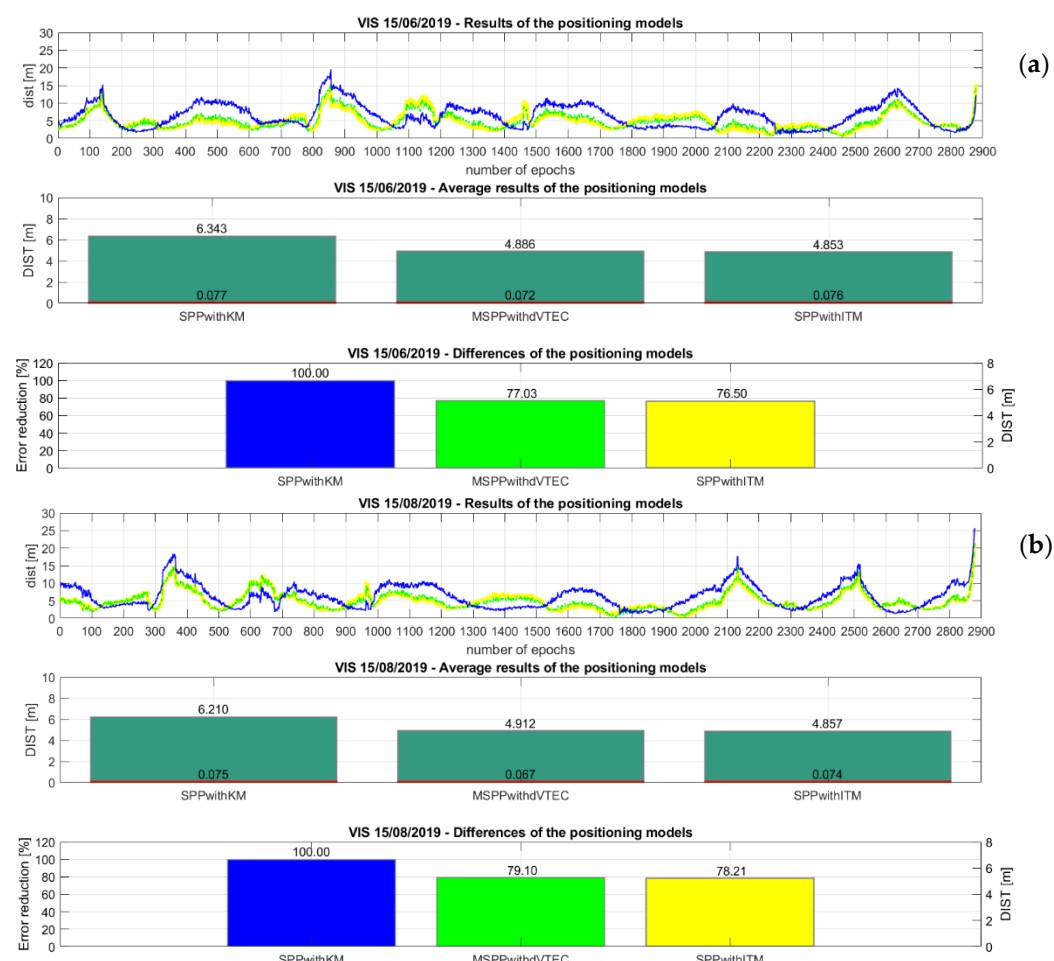

**Figure 1**. Set of the results of the positioning models: (**a**) VIS 15/06/2019 (**b**) VIS 15/08/2019

Following the experiment report, the next examined subject is SKE 15/06/2019. Looking at Figure 2(**a**), the top part presents the *dist* distribution of the **MSPPwithdVTEC** solutions close to the **SPPwithITM**. The average description of *DIST* validates this declaration, where the difference between these two approaches is 0.017 m, in favor of the **MSPPwithdVTEC**. In turn, according to the base model, the **MSPPwithdVTEC** delivers solutions with highly-increased accuracy, which is the most important. Despite such accuracy, the *DIST* precision of **MSPPwithdVTEC** (0.080 m) is improved and is at a similar level as **SPPwithITM** (0.093 m), which confirms the consistency of the methods. Explaining the bottom part of Figure 2(**a**), the error reduction of the **MSPPwithdVTEC** is at the beneficial level of 22.55%, which is again close to the reduction obtained by **SPPwithITM** (22.30%). Therefore, this method can be evaluated as the approach of a similar class compared to the case with IGS TEC map. The second day of tests is 15 August 2019. Based on *dist* in the top of Figure 2(**b**), it is noticeable that the **MSPPwithdVTEC** results are at related relatively similar level as **SPPwithITM.** Looking in the middle part of Figure 2(**b**), the increased accuracy in **MSPPwithdVTEC** is verified by the *DIST* solution

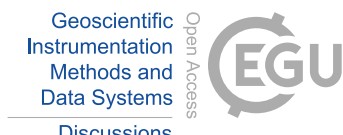

equal to 5.354 m, referring to the initial **SPPwithKM**. The mean error of *DIST* gives an acceptable value using
**MSPPwithdVTEC** by comparable magnitude with the other models. Considering the bottom part of Figure 2(**b**), the error
reduction amounts to 21.30% whereas the approach with the IGS TEC map achieves an equivalent value of 21.07%. In sum,
the **MSPPwithdVTEC** can be assessed on the next EUREF's location as the valuable SPP approach by use of the new
method of the ionospheric refraction estimation, without the need for external products, e.g. atmospheric factors or GIMs.

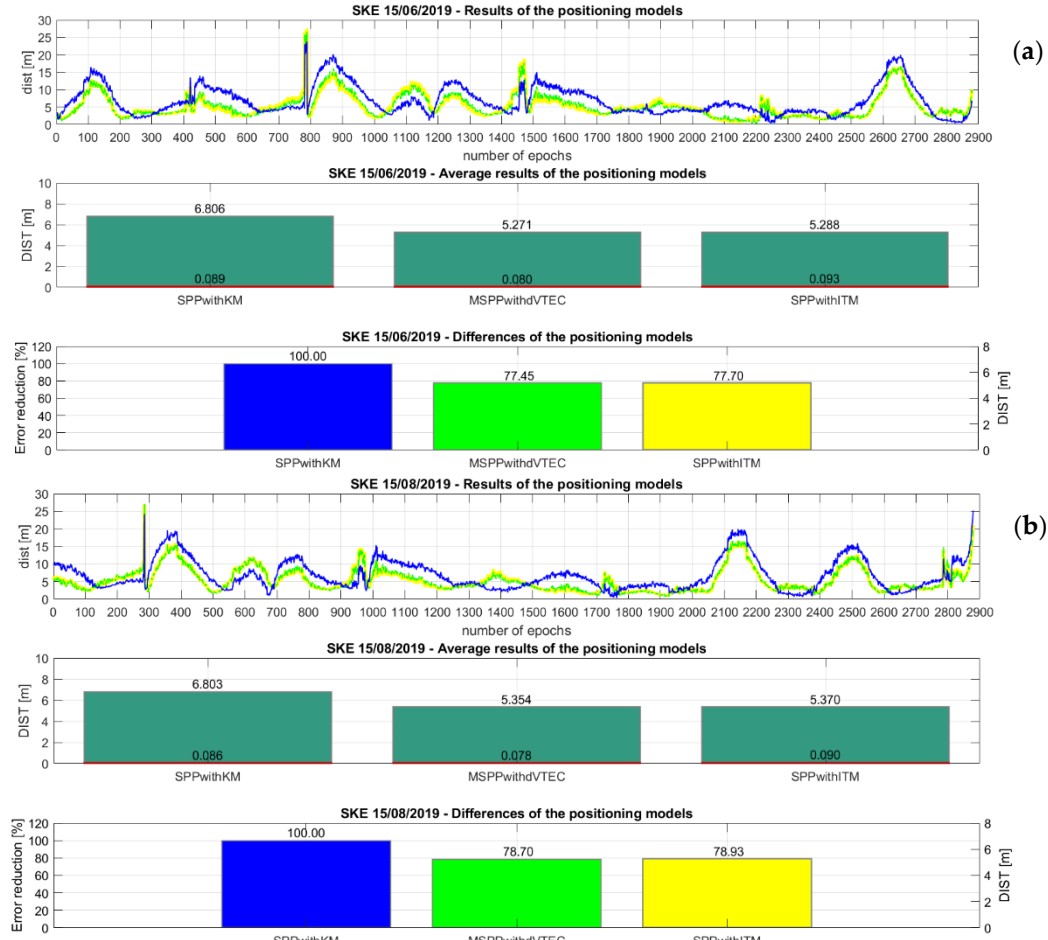

**Figure 2**. Set of the results of the positioning models: (**a**) SKE 15/06/2019 (**b**) SKE 15/08/2019

The last studied point is VARS00NOR. The first examined day is 15 June 2019. The middle part of Figure 3(**a**)

demonstrates that the *DIST* difference of the two approaches: **SPPwithITM** and **MSPPwithdVTEC** is 0.052 m, therefore
the improved accuracy is at a similar level, referring to **SPPwithKM** average observations. The precision of *DIST* confirms
the reliability of the **MSPPwithdVTEC**, where the mean error is equal to 0.087 m with an insignificant discrepancy
(0.008 m) compared to the **SPPwithITM**. The bottom part of Figure 3(**a**) shows a decrease in the percent value of the error.
The error reduction of the **MSPPwithdVTEC** is at the level of 16.69%, thus the improvement of accuracy is verified. Again,





the difference of error reduction among **MSPPwithdVTEC** and **SPPwithITM** is on the parallel level (0.73%) which
confirms the method credibility. The second tested day, and therefore the last one, is 15 August 2019. The *DIST* elaboration
in Figure 3(**b**) presents the low differences between the two principal approaches on the level of 0.028 m. Studying the
bottom division of Figure 3(**b**), the **MSPPwithdVTEC** achieves a positive level of error reduction of 14.91%, relating to the
**SPPwithKM**. In addition, the top parts of Figure 3 (**a**) and (**b**) present the distribution of **MSPPwithdVTEC** *dist* results as
close in value to the **SPPwithITM** with increased accuracy to **SPPwithKM**. This finding is also valid to other examined
cases. Thus, the proposed model can be identified as stable and accurate. The error reduction is at a satisfactory level.

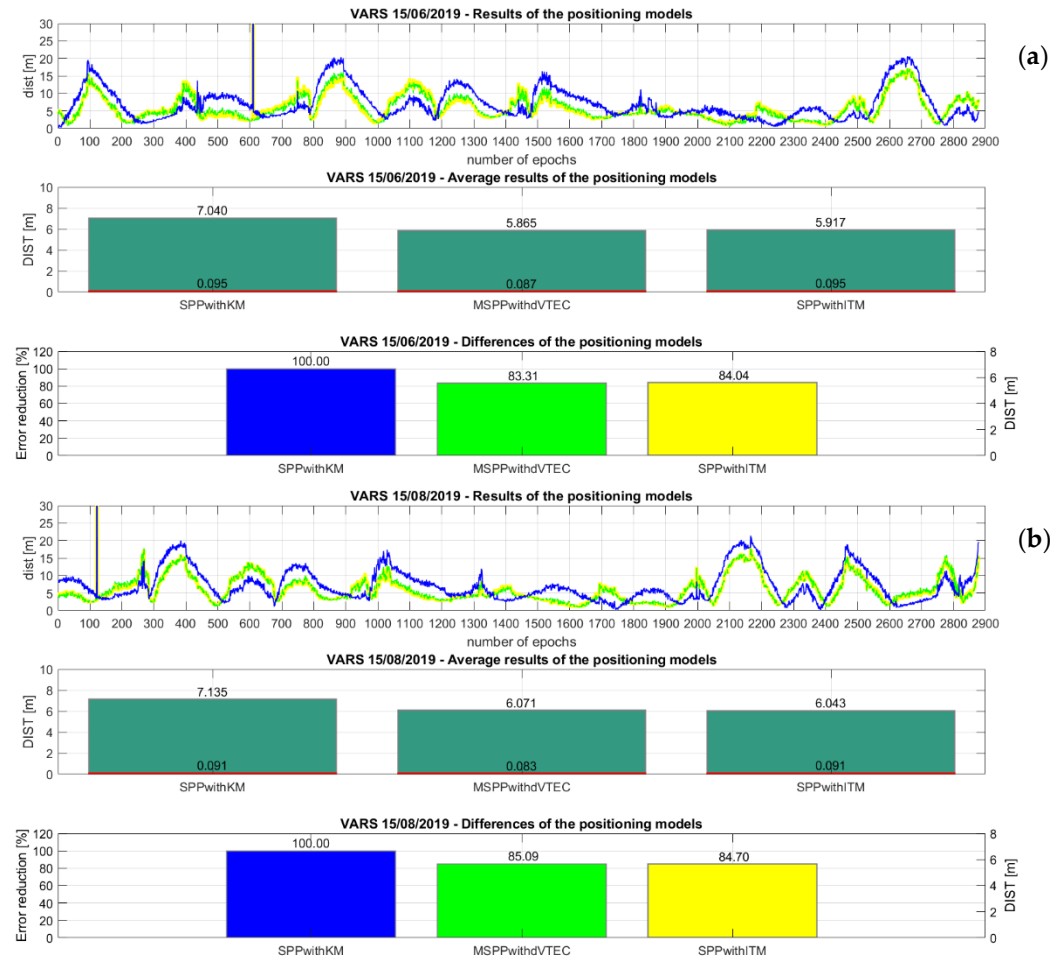

**Figure 3**. Set of the results of the positioning models: (**a**) VARS 15/06/2019 (**b**) VARS 15/08/2019

Focusing on the mean errors of the final solution in the NEU system, we will consider the average precision of the
differences of the components ΔN, ΔE, and ΔU, referring to the daily result. The difference means the discrepancy between
the actual station's coordinates and the received position from the SPP methods. For this purpose, the Eq. (28) was used to
determine the mean values of ΔN, ΔE and ΔU errors which are summarized in the table below:


**Table 3.** Average errors of the difference in the positions using the NEU system

| SPP method | $m_{\Delta N}$ | $m_{\Delta E}$ | $m_{\Delta U}$ | Station and Day |
|---|---|---|---|---|
| SPPwithKM | 0.06 | 0.04 | 0.10 | |
| **MSPPwithdVTEC** | **0.05** | **0.04** | **0.09** | VIS 15/06/2019 |
| SPPwithITM | 0.06 | 0.04 | 0.09 | |
| SPPwithKM | 0.06 | 0.04 | 0.09 | |
| **MSPPwithdVTEC** | **0.06** | **0.03** | **0.08** | VIS 15/08/2019 |
| SPPwithITM | 0.06 | 0.04 | 0.09 | |
| SPPwithKM | 0.05 | 0.03 | 0.11 | |
| **MSPPwithdVTEC** | **0.04** | **0.03** | **0.10** | SKE 15/06/2019 |
| SPPwithITM | 0.05 | 0.03 | 0.11 | |
| SPPwithKM | 0.05 | 0.03 | 0.11 | |
| **MSPPwithdVTEC** | **0.04** | **0.03** | **0.10** | SKE 15/08/2019 |
| SPPwithITM | 0.05 | 0.03 | 0.11 | |
| SPPwithKM | 0.04 | 0.03 | 0.11 | |
| **MSPPwithdVTEC** | **0.04** | **0.03** | **0.10** | VARS 15/06/2019 |
| SPPwithITM | 0.04 | 0.03 | 0.11 | |
| SPPwithKM | 0.04 | 0.03 | 0.11 | |
| **MSPPwithdVTEC** | **0.04** | **0.03** | **0.10** | VARS 15/08/2019 |
| SPPwithITM | 0.04 | 0.03 | 0.11 | |

The error quantities of the difference in the positions were achieved for **MSPPwithdVTEC** and **SPPwithITM** on a close level**.** Separating the horizontal and the vertical components of the position, the **MSPPwithdVTEC** is characterized by improved precision compared to **SPPwithKM** in the North and East direction, therefore, the additional estimated parameter in the code equation does not change the SPP model enough to reduce its quality. The case is repeated in the context of the vertical component U, the **MSPPwithdVTEC** is again profitable to **SPPwithKM** and achieves the similar values of the mean errors to **SPPwithITM**. In general, the values of mean errors are close to each other and the differences are not as clear in the context of the code data use. Therefore, the quantities of average errors demonstrate that **MSPPwithdVTEC** is the approach of the closest precision to the SPP method with an IGS TEC map, specified as a high-quality product, which is the most important from the authors' point of view.

**4 Conclusions and future perspectives**

The main idea of this paper was to introduce the new method to estimate the ionospheric delay in the SPP without using the external data. Moreover, in the case of comparative analysis, two common approaches in SPP was employed: SPP with Klobuchar algorithm and SPP with IGS TEC map. The first one was treated as a reference one. The SPP model with IGS TEC map was utilized to authenticate the proposed model in view of IGS TEC map use - defined as a high-quality product. The explanation of mathematical models and appropriate accuracy analysis criteria was done. Next, the numerical experiment using real code data from three different GNSS stations with discussion to interpret the obtained results.





Referring to achieved solutions, the proposed approach can be defined as a simple and independent way to improve SPP.
Moreover, the **MSPPwithdVTEC** can be employed in the procedure of determination the approximate position for the need
of the single-epoch precise positioning.

Based on the mean distance of the solution from the true position, the **MSPPwithdVTEC** achieved improved GPS

position in comparison to the basic **SPPwithKM** in each tested station. Moreover, the **MSPPwithdVTEC** acquires a similar
level of error reduction to the **SPPwithITM** what is the most satisfying in view of method authentication.

Finally, the results of the **MSPPwithdVTEC** confirm the potential use of the mathematical model in the SPP. The

strategy should be developed in the future through the verification of model stability in the other stations since ionosphere
changes are highly dependent on localization. Therefore, the proposed method of SPP can be recognized as a good forecast
to become independent of external products delivering information about the ionospheric delay.
*Competing interests.* The authors declare that they have no conflict of interest.
*Founding.* This research is supported by grant No. 2018/31/B/ST10/00262 from the Polish National Science Centre.

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
