# Peer review of "estimation based on single epoch data"

_Geoscientific Instrumentation, Methods and Data Systems, 2020_

## Referee Comment (RC1) · Anonymous Referee #1 · 1 Oct 2020

Review of the manuscript: " Single Point Positioning with Vertical Total Electron Content estimation based on single epoch data " by Artur Fischer, Sławomir Cellmer, and Krzysztof Nowel submitted to Geoscientific Instrumentation, Methods and Data Systems.

The reviewed manuscript in general is interesting and well-motivated. It presents a novel approach of ionospheric delay estimation in single point positioning. Undoubtedly, this issue is a current problem described in top scientific journals related to the broadly understood geosciences. For this reason I believe that this manuscript should be considered as a potential publication in the Geoscientific Instrumentation, Methods and Data Systems. Nevertheless, before accepting the manuscript for publication, the authors should make a few minor corrections and respond to the reviewer's questions:

[Figure]

1. Page 1, Line 21 – "Single point positioning (SPP) allows of the indication of an autonomous position of a receiver using code data from the Global Positioning System (GPS)." Does the SPP positioning technique concerns only the GPS system or also other GNSS systems? 2. Equation (9). Not all values in the formula have been explained in the text of the article. 3. Equation (14). If V, X and L are vectors so why then they are written in capital letters? 4. Equation (16). The paper only presents the method of calculating the "mapcoeff" coefficients. Please provide information about determination of the rest of coefficients of matrix A. 5. Equations (20, 21). Least squares solution is widely known in geosciences and I think it can be omitted from the article. 6. Equation (22) . It must be underlined that this formula of the Euclidean distance between points in 3D space is the basic knowledge, so in my opinion there is no need to write this quantity as a new formula. 7. Equation (24). The quantities in the denominator are not explained in the text. 8. Can the approach described in the paper be generalized by combining of the observations from different GNSS systems? If yes, then how would the computational procedure looks then?

---

## Referee Comment (RC2) · Anonymous Referee #2 · 16 Oct 2020

The paper is devoted to the estimation of an absolute position together with ionospheric corrections. The estimation of ionospheric corrections and the position coordinates in one computation process makes the proposed method independent of external data. This topic is in the scope of the Geoscientific Instrumentation, Methods and Data Systems. The problem is presented clearly in the paper. However, some concerns should be clarified: 1. Equation (7): based on cited literature: Leick et al., 2015, the argument of the trigonometric function is a zenith angle at the piercing point. Is it the case in the proposed algorithm? 2. Equation (16): The authors should give the formulas for calculating the entries of the design matrix A. 3. Equation (14), (15), (17), (18), (20): I suggest to denote vectors with small, bold letters. 4. Equation (19): In my opinion, the weight matrix should be denoted as W instead of P. 5. Equation (24): Consequently,

the name of the gradient vector should be changed. The calculation of the entries of the gradient vector should be explained. 6. The authors should describe the results listed in Table 3 in more detail. I recommend the article for publication after clarification of the remarks mentioned above.

---

## Author Comment (AC1) · 4 Nov 2020

Interactive response to the referee comment (Anonymous Referee #1) on "Single Point Positioning with Vertical Total Electron Content estimation based on single epoch data" by Artur Fischer, Sławomir Cellmer and Krzysztof Nowel. First of all, we appreciate your contribution to improving the manuscript. The remarks were implemented into the new version of the paper. The responses (R) to the questions (Q) and comments (C) are as follows: Q1. Page 1, Line 21 – "Single point positioning (SPP) allows of the indication of an autonomous position of a receiver using code data from the Global Positioning System (GPS)." Does the SPP positioning technique concerns only the GPS system or also other GNSS systems? R1. The SPP positioning technique concerns the GPS and other GNSS systems, e.g., GLONASS, GALILEO, or BeiDou. The idea

was to use the real code data in the numerical experiment from the GPS because of the system's popularity, availability and declared precision comparing to other mentioned GNSS systems. C2. Equation (9). Not all values in the formula have been explained in the text of the article. R2. All symbols are explained in the new version of the manuscript. Q3. Equation (14). If V, X and L are vectors so why then they are written in capital letters? R3. This is our mistake. The symbols of vectors were corrected as the reviewer's suggestion. C4. Equation (16). The paper only presents the method of calculating the "mapcoeff" coefficients. Please provide information about determination of the rest of coefficients of matrix. R4. Information about the determination of the rest of the coefficients of matrix is included in the corrected version of the manuscript. C5. Equations (20, 21). Least squares solution is widely known in geosciences and I think it can be omitted from the article. R5. The formulas of LS solution have been removed in the new, improved version of the manuscript. C6. Equation (22). It must be underlined that this formula of the Euclidean distance between points in 3D space is the basic knowledge, so in my opinion there is no need to write this quantity as a new formula. R6. The terms from formula (22) are applied in formula (24). Therefore we decided to present them explicitly, in spite of the basic knowledge of the Euclidean distance formula. C7. Equation (24). The quantities in the denominator are not explained in the text. R7. The terms of equation (24) were briefly explained in the revised version of the research paper. Q8. Can the approach described in the paper be generalized by combining of the observations from different GNSS systems? If yes, then how would the computational procedure looks then? R8. Yes, the approach described in the paper can be generalized by combining the observations from different GNSS systems. First of all, the appropriate observational data from other GNSS systems have to be derived. Taking into account equations (8) and (10), the various variants of them should be considered because of different carrier frequencies of satellite signals from utilized GNSS systems as well as equation (13), related to "mapcoeff" calculation component. The equation (4) will remain unchanged. The mapping function as well, due to subsequent satellites zenith angles of different GNSS systems at the piercing point. Note,

that the value of VTEC0 would be still equal to 5 TECU. Undoubtedly, this quantity can be changed during another experiment containing more than one GNSS system. Taking into account equations (9) and (11), the existed set of GPS code equations should be supplemented by the new observation formulas using satellites of subsequent GNSS systems. The appropriate ingredients of mentioned formulations should be generated in relation to available satellites of GNSS systems, e.g. tropospheric or ionospheric correction components. Afterward, the matrix notation (14) will be updated by the magnification of matrices and vectors structure due to additional observations from following GNSS systems. Nevertheless, the combination of observations from different GNSS systems can be considered as an interesting idea of the next numerical experiment to verify the reliability of SPP with autonomous method of ionospheric delay estimation.

---

## Author Comment (AC2) · 4 Nov 2020

Interactive response to the referee comment (Anonymous Referee #2) on "Single Point Positioning with Vertical Total Electron Content estimation based on single epoch data" by Artur Fischer, Sławomir Cellmer and Krzysztof Nowel. First of all, we appreciate your contribution to improving the manuscript. The remarks were implemented into the new version of the paper. The responses (R) to the questions (Q) and comments (C) are as follows: Q1. Equation (7): based on cited literature: Leick et al., 2015, the argument of the trigonometric function is a zenith angle at the piercing point. Is it the case in the proposed algorithm? R1. This is our mistake. The cited algorithm is referred to the zenith angle calculation at the observing site (from observer's view). The corrected formula was made in the updated version of the manuscript. That was a separate mistake

in the quoted algorithm because the formula used in the code in the MATLAB environment during the numerical experiment was implemented correctly. C2. Equation (16): The authors should give the formulas for calculating the entries of the design matrix A. R2. The formulas for calculating the individual components of the design matrix A were implemented in the revised form of the manuscript. C3. Equation (14), (15), (17), (18), (20): I suggest to denote vectors with small, bold letters. R3. The suggested matrix designations were done in the improved version of scientific research. C4. Equation (19): In my opinion, the weight matrix should be denoted as W instead of P. R4. The recommended symbol for the weight matrix was changed as the reviewer's suggestion. C5. Equation (24): Consequently, the name of the gradient vector should be changed. The calculation of the entries of the gradient vector should be explained. R5. The consequence of the weight matrix designation change is the need for the gradient name change as well. Therefore, the remark was positively included in the corrected version of the manuscript. C6. The authors should describe the results listed in Table 3 in more detail. R6. Basically, the results contained in Table 3 are quite close. Hence, we decided to make the extended description according to the reviewer's suggestion, including the similarity of the values of mean errors of coordinates differences in the NEU system on the comparable level.

---

## Author Response (AR1)

Olsztyn, 06.11.2020

Dear Lev Eppelbaum (levap@post.tau.ac.il)

We appreciate your help in editing the work. We would like to present the response to reviewers' questions and comments as well as marked-up changes (red color into the text) because of the review process which we had to make to improve the paper.

The changes which were made in view of Anonymous Referee #1 report – First Reviewer:

| Records in the old version of manuscript (GI Discussion) / Records with changes in the revised manuscript | Reviewer's questions (Q) and comments (C) / Authors' responses (R) |
|---|---|
| 21-22 | **Q1.** Page 1, Line 21 – "Single point positioning (SPP) allows of the indication of an autonomous position of a receiver using code data from the Global Positioning System (GPS)." Does the SPP positioning technique concerns only the GPS system or also other GNSS systems? |
| 21 - 22 | **R1.** The SPP positioning technique concerns the GPS and other GNSS systems, e.g., GLONASS, GALILEO, or BeiDou. The idea was to use the real code data in the numerical experiment from the GPS because of the system's popularity, availability, and declared precision comparing to other mentioned GNSS systems. |
| 149-159 | **C2.** Equation (9). Not all values in the formula have been explained in the text of the article. |
| 150 - 161 | **R2.** All symbols are explained in the new version of manuscript. |
| 183 | **Q3.** Equation (14). If V, X and L are vectors so why then they are written in capital letters? |
| 183 (183 – 203) | **R3.** This is our mistake. The symbols of vectors were corrected as the reviewer's suggestion. |
| 187 | **C4.** Equation (16). The paper only presents the method of calculating the "mapcoeff" coefficients. Please provide information about determination of the rest of coefficients of matrix. |
| 187 - 191 | **R4.** Information about determination of the rest of coefficients of matrix is included in the corrected version of manuscript. |
| 198-201 | **C5.** Equations (20, 21). Least squares solution is widely known in geosciences and I think it can be omitted from the article. |
| In the improved paper equations do not exist. | **R5.** The formulas of LS solution have been removed in the new, improved version of manuscript. |
| 211 | **C6.** Equation (22) . It must be underlined that this formula of the Euclidean distance between points in 3D space is the basic knowledge, so in my opinion there is no need to write this quantity as a new formula. |
| 209 - 217 | **R6.** The terms from formula (22) are applied in formula (24). Therefore we decided to present them explicitly, in spite of the basic knowledge of the Euclidean distance formula.
 Note, the number of mentioned equations (22) and (24) are in the revised version of manuscript as follows: (20) and (22). |
| 216 | **C7.** Equation (24). The quantities in the denominator are not explained in the text. |
| 214 - 217 | **R7.** The terms of equation (24) were briefly explained in the revised version of the research paper.
 Note, the number of cited equation in the revised form of paper is (22). |
| - | **Q8.** Can the approach described in the paper be generalized by combining of the observations from different GNSS systems? If yes, then how would the computational procedure looks then? |
| - | **R8.** Yes, the approach described in the paper can be generalized by combining of the observations from different GNSS systems. First of all, the appropriate observational data from other GNSS systems have to be derived. Taking into account equations (8) and (10), the various variants of them should be considered because of different carrier frequencies of satellite signals from utilized GNSS systems as well as |

equation (13), related to *mapcoeff* calculation component. The equation (4) will remain unchanged. The mapping function as well, due to subsequent satellites zenith angles of different GNSS systems at the piercing point. Note, that the value of *VTEC$_0$* would be still equal to 5 TECU. Undoubtedly, this quantity can be changed during another experiment containing more than one GNSS system. Taking into account equations (9) and (11), the existed set of GPS code equations should be supplemented by the new observation formulas using satellites of subsequent GNSS systems. The appropriate ingredients of mentioned formulations should be generated in relation to available satellites of GNSS systems, e.g. tropospheric or ionospheric correction components. Afterward, the matrix notation (14) will be updated by magnification of matrices and vectors structure due to additional observations from following GNSS systems.

Nevertheless, the combination of observations from different GNSS systems can be considered as an interested idea of the next numerical experiment to verify the reliability of SPP with autonomous method of ionospheric delay estimation.

The changes which were made in view of Anonymous Referee #2 report – Second Reviewer:

| Records in old version of manuscript (GI Discussion) | Reviewer's questions (Q) and comments (C) |
|---|---|
| Records with changes in the revised manuscript | Authors' responses (R) |
| 145 | **Q1.** Equation (7): based on cited literature: Leick et al., 2015, the argument of the trigonometric function is a zenith angle at the piercing point. Is it the case in the proposed algorithm? |
| 146 | **R1.** This is our mistake. The cited algorithm is referred to the zenith angle calculation at the observing site (from observer's view). The corrected formula was made in the updated version of manuscript. That was a separate mistake in the quoted algorithm because the formula used in the code in the MATLAB environment during numerical experiment was implemented correctly. |
| 187 | **C2.** Equation (16): The authors should give the formulas for calculating the entries of the design matrix A. |
| 187 - 191 | **R2.** The formulas for calculating the individual components of the design matrix A were implemented in the revised form of manuscript. |
| 183 - 204 | **C3.** Equation (14), (15), (17), (18), (20): I suggest to denote vectors with small, bold letters. |
| 183 (183 – 203) | **R3.** The suggested matrix designations were done in the improved version of scientific research. |
| 197 | **C4.** Equation (19): In my opinion, the weight matrix should be denoted as W instead of P. |
| 200 | **R4.** The recommended symbol for the weight matrix was changed as the reviewer's suggestion. (In addition, the names of matrices were changed in view of better understanding:
V => e – theoretical correction,
A – matrix of coefficients without changes,
x – vector of unknown parameters,
$C_x$ – covariance matrix without change,
$m_0^2$ – variance factor without change as well). |
| 216 | **C5.** Equation (24): Consequently, the name of the gradient vector should be changed. The calculation of the entries of the gradient vector should be explained. |
| 215 | **R5.** The consequence of the weight matrix designation change is the need for the gradient name change as well. Therefore, the remark was positively included in the corrected version of the manuscript. Note, in the revised version, the number of cited equation is (22). |
| 369 - 379 | **C6.** The authors should describe the results listed in Table 3 in more detail. |
| 372 - 381 | **R6.** Basically, the results contained in Table 3 are quite close. Hence, we decided to make the extended description according to the reviewer's suggestion, including the similarity of the values of mean errors of coordinates differences in the NEU system on the comparable level. |

[revised manuscript text omitted]
} = -\frac{X^i - X_{r_o}}{\rho^i_{r_o}}, a_{i2} = -\frac{Y^i - Y_{r_o}}{\rho^i_{r_o}}, a_{i3} = -\frac{Z^i - Z_{r_o}}{\rho^i_{r_o}}$ , respectively.  The last column in the first block relates to the clock error in meters.

The vector of unknowns receives an additional parameter in the adjustment process:

$$\mathbf{x} = \begin{bmatrix} \Delta X_r \\ \Delta Y_r \\ \Delta Z_r \\ c\Delta t_r \\ \hline \Delta VTEC \end{bmatrix}$$
(17)

The disclosure vector is:

$$\mathbf{y} = \begin{bmatrix} y_r^1 \\ \vdots \\ y_r^n \\ \hline 0 \end{bmatrix}$$
(18)

where $y_r^i = P_i - \rho_i^r + c\Delta t^i - \delta_{TROP_i} - mapcoeff \cdot VTEC_0$ . The last entry amounts to zero because of assumption (12).

The weight matrix has been prepared based on pseudo-range measurement error which was assumed as a 2.00 m and appropriate satellite elevation angle. The criterion of the minimal mask was implemented as a 10 degree. After computational tests with theoretical analysis, the weight of the estimated component $\Delta VTEC$ was assumed in the model as 1.

$$\mathbf{W} = \begin{bmatrix} \dfrac{1}{\delta^2}\sin(elev_1) & \cdots & 0 & 0 \\ \vdots & \ddots & \vdots & \vdots \\ 0 & \cdots & \dfrac{1}{\delta^2}\sin(elev_n) & 0 \\ \hline 0 & \cdots & 0 & 1 \end{bmatrix}$$
(19)

The least-squares estimate of the equation (14) is computed from the normal equations together with its covariance matrix with the variance factor: $m_0^2 = \dfrac{\mathbf{e^T W e}}{n - m}$ . The number of parameters $m = 5$. Thus, the minimal number of observations should be $n = 6$ to ensure necessary redundancy.

**2.3 Accuracy analysis criteria**

The basic statistical operator in the experiment is a distance of the solution from the true position *dist* where subscript

"*r*" means calculated rover's coordinates and "*t*" regarding to the actual position. Moreover, its average value (*DIST*), computed from solutions obtained from the single epochs with its mean error. The actual position means constant station coordinates provided by the agency, which manage the Continuously Operating Reference Station (CORS) used in the experiment for evaluation of the positioning model accuracy. The formula can be introduced in each epoch in the form of

Euclidean distance:

$$dist_{ep_i} = \sqrt{(X_r - X_t)^2 + (Y_r - Y_t)^2 + (Z_r - Z_t)^2} \tag{20}$$

The formula to calculate the mean error of average solution is as follows:

$$m^2_{dist_{ep_i}} = \mathbf{G} \mathbf{C}_{\hat{\mathbf{X}}_{ep1}} \mathbf{G}' \tag{21}$$

where $\mathbf{C}_{\hat{\mathbf{x}}}$ is a covariance matrix of the parameter vector and $\mathbf{G}$ is a gradient:

$$\mathbf{G} = \begin{bmatrix} \dfrac{\Delta X_{ep_i}}{dist_{ep_i}} & \dfrac{\Delta Y_{ep_i}}{dist_{ep_i}} & \dfrac{\Delta Z_{ep_i}}{dist_{ep_i}} \end{bmatrix} \tag{22}$$

[revised manuscript text omitted]